# Improved Immunogenicity of the Inactivated F Genotype Mumps Vaccine against Diverse Circulating Mumps Viruses in Mice

**DOI:** 10.3390/vaccines11010106

**Published:** 2023-01-01

**Authors:** Hyeran Won, Ah-Ra Kim, Gyung Tae Chung, Su Hwan Kim, Jung-Sik Yoo, June-Woo Lee

**Affiliations:** 1Division of Vaccine Research, Center for Vaccine Research, Korea National Institute of Infectious Diseases, Cheongju 28160, Chungcheongbuk-do, Republic of Korea; 2Division of Vaccine Clinical Research, Center for Vaccine Research, Korea National Institute of Infectious Diseases, Cheongju 28160, Chungcheongbuk-do, Republic of Korea; 3Center for Infectious Disease, National Institute of Health, Korea Disease Control and Prevention Agency, Cheongju 28159, Chungcheongbuk-do, Republic of Korea; 4Division of Antimicrobial Resistance, National Institute of Health, Korea Disease Control and Prevention Agency, Cheongju 28159, Chungcheongbuk-do, Republic of Korea

**Keywords:** mumps virus, formalin, vaccine, immunogenicity

## Abstract

Mumps is an acute infectious disease caused by the mumps virus (MuV). Despite high global vaccination coverage, mumps outbreaks continue to occur, even in vaccinated populations. Therefore, we aimed to identify candidate vaccines that can induce an immunogenic response against diverse MuV genotypes with greater efficacy than the currently available options. Vaccine candidates were sourced using formalin-inactivated viral strains. The inactivated vaccines were administered to BALB/c mice (through a primer and booster dose administered after a three-week interval). We tested the neutralizing antibodies of the candidate vaccines against various MuV genotypes to determine their overall efficacy. The formalin-inactivated F genotype vaccine was found to have higher cross-neutralizing titers against genotypes F, H, and G as well as significant Th1 cytokines responses, IFN-γ, TNF-α, and IL-2 than the Jeryl Lynn (JL) vaccine. Our findings suggest that the inactivated F genotype mumps vaccine has higher immunogenicity than the JL vaccine against diverse circulating MuVs.

## 1. Introduction

Mumps is an acute viral infection that is caused by the mumps virus (MuV) [1]. MuV harbors a non-segmented, single-stranded RNA genome (15,384-nucleotide long) encoding the nucleocapsid (N), phospho (P), matrix (M), fusion (F), small hydrophobic (SH), hemagglutinin-neuraminidase (HN), and large (L) proteins [1,2]. The World Health Organization (WHO) has standardized a system for MuV genotyping, wherein there are 12 MuV genotypes (from A to N, excluding E and M) based on the nucleotide sequences of the gene encoding SH—the most variable genomic region—and HN.

Seroprevalence studies have shown that the efficacy of the mumps vaccine ranges from 66% to 95% with a median of 88%, whereas that of measles and rubella is at least 94% [3]. Additionally, the seropositivity against MuV is lower than that against measles or rubella viruses among MMR vaccine recipients [4,5,6]. Despite the availability of the mumps vaccine, and its high population coverage, mumps outbreaks continue to occur in vaccinated populations worldwide [7,8,9].

Many factors have been suggested as possible causes of these mumps outbreaks, including the waning of immunity with time and antigenic mismatch of vaccine strain to the wild type virus [10]. As immunity against mumps weakens with the passage of time, lack of booster vaccine use may be a potential factor contributing to mumps resurgence and large global outbreaks. This is a reasonable possibility as most mumps cases and outbreaks are known to occur among young adults [9,11]. Although MuV is serologically monotypic, 12 genotypes show different antigenicity between genotypes. It elicits the different level of cross-neutralizing antibodies against genetically different MuV strains [12], thus antigenic mismatch may further contribute to mumps outbreaks in vaccinated populations [13,14,15,16,17,18]. Evolutionary analysis of outbreak strains in the United States between 2016 and 2017 implicated a single viral lineage that could be traced back to the outbreaks in 2006; It is thought that continuous transmission and unreported cases are key factors for disease persistence and recurrent outbreak [19].

If genetic variations persist in viral genome by natural selection, cross-reactivity induced vaccine will continue to decrease.

Diverse platforms of vaccine candidates have been studied to improve mumps vaccine due to continual resurgence. Synthetic DNA vaccine targeting the HN or F protein of MuV was studied [20], and genetically modified DNA vaccines lacking the genes encoding SH (rMuVΔSH), V gene (rMuVΔV), or both (rMuVΔSHΔV), have also been developed. In addition, a full-length cDNA clone of recombinant MuV-S79 has been evaluated as a potential vaccine [21]. The live attenuated mumps vaccine, the currently used vaccine platform, was subjected to empirical attenuation by the repeated passage of the F genotype virus in cultured cells. The replacement of the A genotype strain with the F genotype MuV, was not inferior to the A genotype vaccine [22,23]. The high immunogenicity of attenuated vaccine candidates using MuV of the F genotype indicates that the F genotype is a candidate genotype to replace vaccine strains.

Inactivated vaccines have been widely used over the past 70 years due to their safety, and ease of storage, and transportation and are used for prevention of infectious diseases, such as hepatitis A, polio, influenza, and SARS-CoV-2 [24]. Formalin is commonly used for viral inactivation during the manufacture of vaccines against viruses such as the hepatitis A virus, polio, influenza virus, human papillomavirus, and hantavirus [25,26,27,28,29]. The value of formalin lies in its efficacy as an alkylating agent, which affects both the viral genome and proteome; it can crosslink RNA to the capsid proteins to block transcription, and it can also crosslink methylene bridges between primary amino groups to prevent the transcriptional machinery from accessing the genome. These changes can attenuate the ability of the infectious agent to replicate within the body, while also maintaining the integrity of diverse antigenic epitopes to support the development of an effective immune response against the pathogen [29].

CD4+ T helper (TH) or CD8+ cytotoxic T cells (CTLs) play an important role in the elimination of MuV-infected cells [30,31,32]. MuV infection–rather than childhood vaccination induces persistent polyfunctional CD8 T-cell memory [33]. However, little is known about cell-mediated immunity in response to MMR vaccination. In this study, we evaluated the immunogenicity of inactivated F genotype mumps vaccine and compared it with that of the Jeryl Lynn vaccine in mice. Inactivated F genotype mumps vaccine could provide broad-spectrum protection against diverse genotypes of mumps viruses in humoral and cellular immunity.

## 2. Materials and Methods

### 2.1. Cell Culture

Vero cells (ATCC: CCL-81) were maintained in Dulbecco’s modified minimal essential medium (DMEM; 1995065, Gibco) supplemented with 10% heat-inactivated fetal bovine serum (FBS; GIB-10082-147, Gibco) and 1× penicillin-streptomycin (P/S; GIB-15140-122, Gibco) at 37 °C with 5% CO_2_.

### 2.2. Virus Strains

We obtained the JL mumps virus vaccine strain from the Korean Ministry of Food and Drug Safety (No. 0666296). MuVi/Incheon.KOR/16.08/22[F] (genotype F), MuVi/Gyeonggi.KOR/26.16/1[H] (genotype H), and MuVi/Jeonnam.KOR/10.15/5[I] (genotype I) were isolated from throat swabs and saliva (from patients). MuVi/Iowa.US/2006[G] (genotype G) was from ATCC (VR-1899). MuVs were propagated in Vero cells at a multiplicity of infection (MOI) of 0.01 for 48 h in minimum essential medium (MEM) supplemented with 2% FBS and 1% penicillin-streptomycin at 37 °C with 5% CO_2_. The MuV-infected cells were harvested when a cytopathic effect (CPE) of 70%–80% was reached, following repeated freeze–thaw cycles.

### 2.3. Virus Titrations

The concentration of the infectious virus particles was determined using a plaque-forming assay or a focus-forming assay.

#### 2.3.1. Plaque Assay

Vero cells were seeded at a density of 2 × 10^5^ per well in 12-well plates containing DMEM supplemented with 10% FBS and 1× penicillin-streptomycin and then incubated at 37 °C in 5% CO_2_ for 24 h. Then, the cells were infected with serially diluted stock virus prepared in MEM at 37 °C in 5% CO_2_. After 2 h, the inoculum was removed, and the cell monolayer was overlaid with plaquing media prepared from 2× MEM supplemented with 4% FBS, 2% P/S and autoclaved 2% low melting agarose with distilled water in a sterile bottle at a 1:1 ratio. This culture was incubated at 37 °C in 5% CO_2_ for 72 h while monitoring the appearance of plaques. Upon confirmation of visible plaques, the cells were fixed in 4% formaldehyde in phosphate-buffered saline (PBS) and stained with 1% crystal violet solution. Viral titers were calculated in terms of the plaque-forming units per mL (PFU/mL).

#### 2.3.2. Focus Forming Assay

Focus forming assay was performed according to a previously described method with Vero cells [18]. Viral titers were calculated in focus forming units per mL (FFU/mL).

### 2.4. Purification and Inactivation of Candidate Vaccines

Viral supernatants were harvested at 3–4 days post-infection (dpi), subjected to centrifugation at 3000 rpm for 10 min at 4 °C, and filtered through a 0.8-μm filter to remove the cell debris. The clarified viral supernatants (200 mL each candidate) were rapidly thawed in a 37 °C water bath and treated with 20 U/mL of benzonase (Merck Millipore, Waltham, MA, USA) overnight at 4 °C to digest host cell DNA, followed by inactivated in 0.05% formalin (*v*/*v*) (Sigma Aldrich, Burlington, MA, USA) for 2 days at 4 °C and centrifuged at 25,000 rpm at 4 °C for 2 h [SW32Ti rotor (Beckman Counter)]. Pellets were resuspended in PBS and concentrated in a 15% sucrose (*w*/*v*) cushion at 25,000 rpm and 4 °C for 2 h. Candidate vaccines were then diluted with PBS, and protein concentrations were determined using Bradford Protein Assay (Coomassie-Bradford protein assay kit, ThermoFisher Scientific, Waltham, MA, USA).

### 2.5. Animals and Vaccination Schedule

Female BALB/c mice aged 4 weeks were divided into three groups, i.e., the experimental group, positive control, and negative control. The experimental group received an intramuscular (i.m.) injection of 10 µg inactivated mumps virus (JL strain genotype A; wild-type MuVs; F, H, I, or G) at 2.5 × 10^5^ ffu; the virus was administered through two intramuscular (i.m.) injections (a primer and a booster after a three-week interval). (Figure 1A) Using the same vaccination schedule, the positive and negative control groups were administered live MuV and PBS, respectively. For serological analysis, blood samples were obtained through a retro-orbital plexus puncture 3, 6, and 9 weeks after administering the primer; the first sample was obtained before injecting the booster dose. The sample from week 9 was used to assess antibody duration. Figure 1 and Figure 2A,B were performed with 5 mice, and Figure 2C was performed with 3 mice. Figure 3 was performed on 10 mice, all individually.

### 2.6. Immunoblot Analysis

Protein concentration was quantified using the Bradford protein assay (Pierce Coomassie protein assay kit, ThermoFisher Scientific, Waltham, MA, USA). Samples (20 μg of each) separated on 4%–12% NuPage polyacrylamide gels (ThermoFisher Scientific, Waltham, MA, USA) and transferred onto nitrocellulose membrane, and blocked with 5% milk prepared in PBS/T (0.05% Tween-20) for 30 min. The membranes were then incubated overnight at 4 °C with rabbit anti-mumps antibody (1:1000; MUMNS11-SB; Alpha Diagnostic International, San Antonio, TX, USA). The following day, the membranes were washed with PBS/T (0.05% Tween-20), incubated with goat anti-rabbit IgG (H + L) secondary antibody, and HRP (1:2000; ThermoFisher Scientific, Waltham, MA, USA) for 1 h at 20 °C, and washed again, prior to colorimetric assessment with 4CN Plus Chromogenic (PerkinElmer, Waltham, MA, USA) in accordance with the manufacturer’s protocol. The intensity of relative bands was measured using ImageJ (National Institutes of Health, Bethesda, MD, USA).

### 2.7. Immunologic Assay

#### 2.7.1. Enzyme-Linked Immunosorbent Assay (ELISA)

MuV-specific total IgG antibody titers were determined using ELISA. 50 μL of MuV mixtures (1 × 10^5^ ffu/mL of each genotype A, F, H, I, and G) was added to each well and were coated overnight in 96-well plates at 4 °C, and the mixtures were washed with PBS/T (0.05% *v*/*v*) and blocked with 1% BSA and 0.5% FBS in PBS/T for 2 h at 37 °C. Mouse serum was serially diluted and incubated with these mixtures for 2 h at 37 °C, following which, the mixtures were washed, probed with HRP-conjugated goat anti-mouse IgG secondary antibodies (1:2000) for 1 h at 37 °C, and washed again with PBS/T (0.05% *v*/*v*). Tetramethylbenzidine (TMB) was then added as a substrate, and the reaction was quenched by adding the stop solution for TMB (genDEPOT). For each sample, the optical density (OD) was measured at 450 nm using an ELISA reader (Molecular Device, San Jose, CA, USA, SpectraMax i3x) and correlated to values in the standard curve. MuV-specific total IgG antibody titers were tested in 3 wells per mouse, and the mean values were used for statistical analysis.

#### 2.7.2. Focus-Reduction Neutralization Test (FRNT)

FRNT was performed according to a previously described method with a few modifications [18]. We used a FRNT assay to detect the relative cross-genotype protection provided by the candidate vaccines against JL genotype A, MuVi/Incheon.KOR/16.08/22[F] (genotype F), MuVi/Gyeonggi.KOR/26.16/1[H] (genotype H), and MuVi/Jeonnam.KOR/10.15/5[I] (genotype I) or MuVi/Iowa.US/2006[G] (genotype G). The difference is that we used Vero cells instead of Vero hSLAM cells. Neutralizing antibody titers were tested in 3 wells per mouse, and the mean values were used for statistical analysis.

#### 2.7.3. T-Cell ELISpot Assays

T-cell responses were determined by gamma interferon (IFN-γ) enzyme-linked immunosorbent spot (ELISpot) assay [BD, ELISPOT for mouse IFN-γ, (Interleukin) IL2 or IL4] according to the manufacturer’s protocol. Spleens were harvested from immunized mice after booster immunization. Splenocytes were filtered through a 100-μm pore size nylon cell strainer (BD) and digested with red blood cell lysis buffer (Sigma) to obtain a single-cell suspension. The splenocytes were seeded at a density of 1 × 10^6^ per well in 96-well plates containing RPMI supplemented with 10% FBS and 1× P/S at 37 °C with 5% CO_2_ for 2 h. The splenocytes were stimulated with a mixture of MuVs (1 × 10^5^ ffu/mL of each genotype A, F, H, I, and G) and incubated at 37 °C, 5% CO_2_ for 24 h. After stimulation, the cells were incubated with biotin-conjugated antibodies and streptavidin-HRP. Spots were developed using 3-amino-9 ethylcarbazole (AEC) substrate (BD ELISAPOT AEC substrate set). The number of IFN-γ-, IL2-, or IL4-secreting cells was counted using the automated ELISPOT reader as described above. Data are presented as the number of spot-forming units (SFUs) per 10^6^ splenocytes.

#### 2.7.4. Cytokine Quantification

The levels of cytokine secreted into the supernatants were quantified by multiplex for IFN-γ, IL-2, IL-4, IL-10, and TNF-α (R&D systems LXSAMSM) in accordance with the manufacturer’s instructions. Splenocytes were isolated from the spleens of immunized mice. The splenocytes were seeded at a density of 1 × 10^6^ per well in 96-well plates containing RPMI supplemented with 10% FBS and 1× P/S at 37 °C with 5% CO_2_ for 2 h. The splenocytes were stimulated with a mixture of MuVs (1 × 10^5^ ffu/mL of each genotype A, F, H, I, and G) and incubated at 37 °C, 5% CO_2_ for 24 h. After harvesting the cells and cell culture medium, they were centrifuged, and cytokines were measured in the supernatant. The supernatants were mixed with beads coated with capture antibodies and incubated for 16–18 h at 4 °C. The beads were washed and incubated with biotin-labeled detection antibodies for 1 h at RT and incubated with streptavidin-PE. The median fluorescence intensity of the beads was analyzed using a flow-based reader (Luminex 200) according to the manufacturer’s instructions.

#### 2.7.5. Histology and Immunohistochemistry

Spleen tissue was fixed in 4% paraformaldehyde (PFA) prepared in PBS for 1 d, embedded in paraffin, sectioned at 4-μm thick, and mounted on glass slides. The sections were deparaffinized in xylene, rehydrated in a graded ethanol series, and then stained with hematoxylin and eosin (H&E).

For immunohistochemistry (IHC), the sections were treated with proteinase K solution (0.6 units/mL in TE buffer, pH 8.0) for 5 min at RT. CD3 for T-cells (1:150; clone SP7, Abcam) was used with mouse and rabbit HRP/DAB IHC Detection kit polymer (Abcam) according to the manufacturer’s instructions. The expressions of CD3 were determined by the percent positive staining of at least 3 images. Images were acquired at 400× magnification using an Olympus BX51 microscope and analyzed at 86× using Motix Images plus 3.0.

### 2.8. Phylogenetic Analysis

A phylogenetic tree was generated from the nucleotide ClustalW alignment of the available MuV whole-genome sequences and Korean MuV isolates in BioEdit using the neighbor-joining method with 1000 bootstrap replicates in MEGA 11. The phylogenetic tree is drawn to scale, with branch lengths in the same units as those of the evolutionary distances used to infer the phylogenetic tree in the units of the number of base substitutions per site. This analysis involved 23 nucleotide sequences. All ambiguous positions were removed for each sequence pair (pairwise deletion option). There were a total of 15,349 positions in the final dataset. Phylogenetic trees were further developed from Korean isolates and reference sequences from GenBank based on whole-genome sequences. The differences in the HN and F protein sequences were compared between the JL strain and the wild-type MuVs.

### 2.9. Ethical Statement

This study was ethically approved by the Institutional Animal Care and Use Committee of the Korea Centers for Disease Control and Prevention (approval number: KCDC-069-19-2A, KCDC-035-20-1A). All experiments were conducted in accordance with biosafty Institutional Biosafety Committee (IBC) and approved by the Korea Centers for Disease Control and Prevention (approval number: KCDC-IBC-2018-065).

### 2.10. Statistical Analysis

Data were analyzed using a two-tailed Student’s *t*-test with excel and expressed as the mean ± standard deviation (SD) of independent experiments. *p*-values < 0.05 indicate statistically significant differences.

## 3. Results

### 3.1. Preparation of Inactivated Mumps Vaccine and Humoral Immunity of Inactivated Mumps Vaccine Candidates in BALB/c Mice

To generate an inactivated mumps vaccine, the supernatants containing various [JL (A), F, H, I, and G] genotype were treated with 0.05% formalin for 2 days at 4 °C. After purification and concentration, the degree of completeness of inactivation of each inactivated mumps vaccine candidate was determined using a focus-forming assay (Appendix A). In addition, the candidates were immunized and tested for humoral immunity in 4-week-old BALB/c mice (Appendix A). None of the groups have a higher total IgG response than the JL group. However, the F and G genotypes of the mumps vaccine have high neutralization titers against various genotypes of MuV (Appendix A). Therefore, two candidates were selected for the experiment. The immunization schedules are depicted in Figure 1A. The currently used JL (A) vaccine concentration is approximately 1 × 10^4^ pfu. The viral titers were calculated in plaque-forming units per mL (PFU/mL) and the focus-forming units per mL (FFU/mL) with various mumps viruses. We found that 1 × 10^4^ pfu corresponds to 2.5 × 10^5^ ffu (Data not shown). Therefore, experiments were performed with a vaccination concentration of 2.5 × 10^5^ ffu. ELISA using the MuV-specific serum antibodies revealed that the total IgG response was higher in all immunized groups than in the negative control group, although there were no significant differences among the five tested groups (Figure 1B). FRNT analysis of cross-protection conferred by the vaccine candidates revealed that the neutralizing antibodies against the F, H, and G MuV genotypes were significantly higher in the inactivated F genotype mumps vaccine (F10) immunized group (n = 5) than in the JL group (Figure 1C).

The purity of the inactivated mumps vaccine candidate was assessed by SDS-PAGE. The band intensity of the HN protein, a surface protein of MuV of the inactivated mumps vaccine was increased compared with that of the live MuV. Both the monomeric and dimeric forms of the HN protein were identified at approximately 40 and 170 kDa, respectively (Appendix A).

### 3.2. Cell-Mediated Immunity of Inactivated Mumps Vaccine Candidates in BALB/c Mice

To explore the cellular immune response triggered by the F10 inactivated vaccine, an ELISPOT assay was performed to test IFN-γ, IL-2, and IL-4 using an immunized BALB/c mice splenocytes stimulated with mixtures of MuVs. The levels of IFN-γ and IL-2 were significantly elevated in F10 group compared to the JL group (*p* < 0.001, *p* < 0.05, respectively) (Figure 2A). The multiplex assay of cytokine levels also revealed that IFN-γ, IL-2, and TNF-α levels were higher in F10 group compared to JL group after MuV stimulation (*p* = 0.000009, 0.015, 0.0004, respectively). Moreover, IL-4 and IL-10, were similar in F10-inactivated and JL-immunized mice (Figure 2B).

To examine the activation of cytotoxic T cells and T helper cells, CD3 levels in the spleen were examined. As shown in Figure 2C,D, increased CD3 levels were observed in F10 immunized mice compared with JL immunized mice. A histological analysis of the spleen revealed no significant differences among all groups after vaccination.

### 3.3. Longevity of Humoral Immunity from Inactivated Mumps Vaccine Candidates in BALB/c Mice

To investigate the longevity of humoral immunity from the inactivated mumps vaccine candidates, we assessed cross-neutralization antibody titers by FRNT against diverse MuVs at 6 and 9 weeks after priming. Total IgG response against MuV was significantly higher among all immunized groups as compared with the PBS-immunized group; however, there were no significant differences among the five groups 9 weeks after priming (Appendix A). Similar results were obtained in groups 6 weeks after priming (Figure 1B). Enhanced levels of cross-neutralization antibody titers against the MuV genotypes F, H, and G were observed in the F10 group compared with the JL group (Figure 3). Although inactivated vaccines generally induce antibody protection for a much shorter duration, inactivated F genotype mumps vaccines continuously induced strong humoral immunity up to 9 weeks after the priming vaccination.

### 3.4. Comparison of Antigenic Sites and Whole-Genome Sequences between Genotypes A and F Mumps Virus

A phylogenetic analysis based on whole genome sequences was performed to examine differences between genotype A, vaccine strain, and F: inactivated vaccine candidate. It also included 16 WHO reference strains. There were phylogenetic differences between the two strains (Figure 4A). Furthermore, the N-glycosylation site at amino acid (aa) 464 is absent in the vaccine strain. Of the three neutralizing epitope sites of the HN protein, positions 4–6 were mutated in the MuV genotypes F, H, I, and G compared with the JL vaccine strain. In addition, the fusion promotion site at aa 195 in the F protein of MuV, MuVi/Incheon.KOR/16.08/22[F] is the same as that of the JL vaccine strain, whereas that in the strain SP, which has been previously studied as an attenuated vaccine, has a mutation of S to F (Figure 4B).

## 4. Discussion

Despite high vaccination rates, mumps outbreaks continue to occur. This is considered to be due to a combination of waning immunity and antigenic mismatch, leading to the development of new, more effective mumps vaccines against circulating MuV genotypes.

Although non-infectious, inactivated viruses contain multiple viral proteins for effective immune recognition. Moreover, inactivated vaccines can be easily mass-produced, and have been traditionally used for effective vaccine development, particularly against viral diseases [34]. The formalin-inactivated mumps vaccine and a detergent-inactivated vaccine containing an adjuvant have been evaluated in rhesus macaques and mice, respectively [35,36]. Herein, the inactivated mumps vaccine was developed while considering productivity and safety.

Vaccines typically mediate protection through the induction of pathogen-specific serum antibodies produced by B lymphocytes. Neutralizing antibodies are considered a major correlate of protective immunity for the development of vaccine candidates. We first tested mumps-specific IgG and neutralizing antibody titers in immunized mice to determine the humoral immunity induced by the diverse genotypes of the inactivated mumps vaccine. Figure 1B,C and Figure 3A show the F10 inactivated vaccine immunized mice and elicited mumps-specific IgG antibodies shows higher levels of neutralizing antibodies, which maintained them 9 weeks after priming as compared to JL immunized mice against diverse genotypes of MuV.

Besides humoral immune response, Type 1 T helper (Th1)-biased T-cell responses are important for clearing virus-infected host cells for vaccine development. Cytokines produced by Th cells are vital to host defense against viruses. In addition, Th1 cells promote cell-mediated immune response and produces IFN-γ, IL-2, and tumor necrosis factor (TNF)-α.

The inactivated vaccine candidates are generally formulated with adjuvants to facilitate an immune response. However, the F10 inactivated vaccine induced a more robust humoral and cell-mediated immune response without adjuvants, resulting in greater cost-effectiveness.

In this study, F10-inactivated immunized mice showed higher levels of CD3, as T cell markers in the spleen than JL-immunized mice. The levels of cytokines, such as IFN-γ, IL-2, and TNF-α, were also significantly higher in splenocytes after MuV stimulation in F10-inactivated immunized mice compared with JL immunized mice. This suggests that F10-inactivated vaccine induced faster viral clearance and constrained the spread of the mumps virus. On the other hand, cytokines which are induced by th2 cells, such as IL-4, and IL-10, were similar in F10-inactivated and JL-immunized mice.

Although the F10-inactivated vaccine and the JL vaccine were similar in Th2 cell response, F10 yielded higher neutralizing antibody titers and Th1 cell responses against diverse genotypes of MuVs. Therefore, additional information on cell-mediated immunity is required to holistically evaluate the newly developed mumps vaccines.

Inactivated candidates MuV vaccines have been evaluated for immunogenicity in mice or monkeys [35,36]. However, there is a difference in nucleotide sequences between our vaccine strain and the SP strain (Figure 4). Therefore, it is expected that there will be differences in immunogenicity. Our data showed that the F10 inactivated vaccine candidate has the ability to cross-neutralization against various MuV genotypes, making it useful for a new vaccine strain.

One limitation of this study is that no challenge experiments were undertaken in mice lacking the interferon receptor α/β, which are susceptible to MuV infection [37]. Thus, the efficacy of the F10 inactivated mumps vaccine against mumps infection must be confirmed through further study.

In conclusion, we demonstrate a well-tolerated inactivated mumps vaccine that can stimulate strong humoral and cell-mediated responses in BALB/c mice. Considering that the levels of neutralizing antibodies against diverse MuV genotypes were as high as those of the convalescent sera, these results highlight the safety, immunogenicity, and overall efficacy of potential inactivated vaccine candidates for transitional and early-phase clinical trials.

## Figures and Tables

**Figure 1 vaccines-11-00106-f001:**
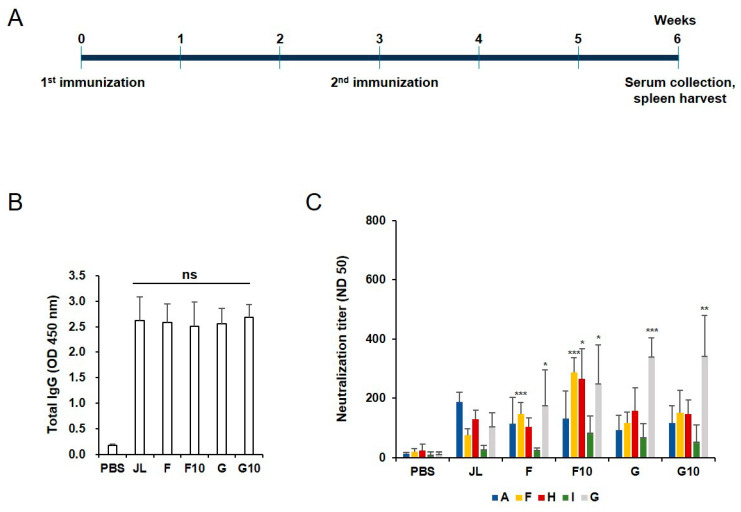
Administration schedule of inactivated mumps vaccine candidates and humoral responses of inactivated mumps vaccine candidates. (**A**) Experimental groups. BALB/c mice were immunized intramuscularly with 10 μg of inactivated mumps vaccine candidates (F10 and G10), 2.5 × 10^5^ ffu/mL of F, G genotypes of MuV, which is the wild-type virus control, and the Jeryl Lynn vaccine (genotype A) which is a positive control and a currently available vaccine at weeks 0 and 3. The blood samples and spleens were collected at week 6. (**B**) Total IgG elicited in BALB/c mice serum among different immune groups measured using ELISA. All vaccinated mice induced a significant MuV-specific IgG response. Not significant (NS) when compared between immunized groups. (**C**) Serum neutralizing antibody responses against diverse genotypes of MuV. Data are presented as the mean ± SD pooled from triplicate experiments (N = 5). Neutralizing antibodies against the F, H, and G MuV genotypes were significantly higher in the F10 immunized group (n = 5) compared to the JL immunized group. * *p* < 0.05, ** *p* < 0.01, *** *p* < 0.005 analyzed by unpaired *t*-test.

**Figure 2 vaccines-11-00106-f002:**
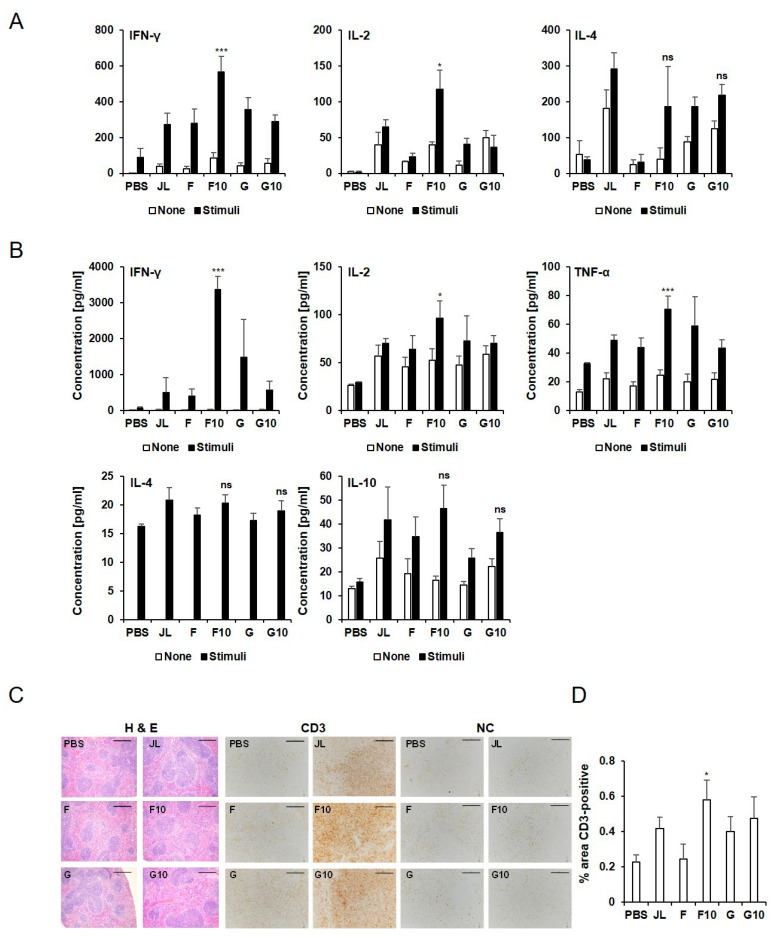
Cellular responses of inactivated mumps vaccine candidates. (**A**) IFN-γ, IL-2, and IL-4 ELISPOT analysis was performed, and the spots were counted. (**B**) Secreted IFN- γ, IL-2 TNF-α, IL-4, and IL-10 levels were detected by multiplex. Data are presented as the mean ± SD pooled from individual mice (N = 5). Th1 cytokine responses were significantly increased in the F10 immunized group (n = 5) compared to the JL immunized group. * *p* < 0.05, *** *p* < 0.005 analyzed by unpaired *t*-test. (**C**) CD3 levels were examined by IHC in spleen (N = 3). Representative staining images of the splenic history of immunized mice. (**D**) Percentage of immunostained area for CD3. Scale bar: 100 μm.

**Figure 3 vaccines-11-00106-f003:**
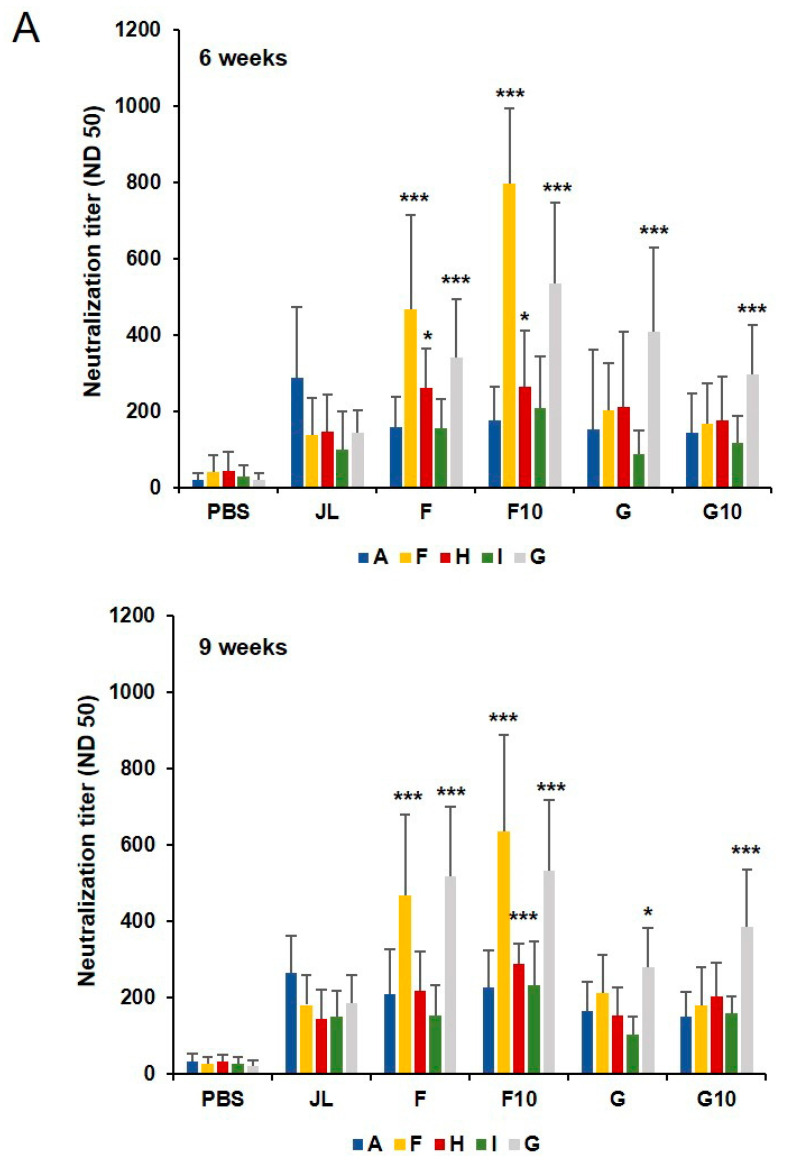
Longevity of humoral immunity of inactivated mumps vaccine candidates. (**A**) Serum-neutralizing antibody responses against diverse genotypes of MuV at 6 and 9 weeks. The average number of spots was calculated in triplicate. The results of 50% neutralizing antibody titers are presented as the mean values from individual mice (N = 10). Neutralizing antibodies against the F, H, and G MuV genotypes were significantly higher in the F10 immunized group (n = 5) compared to the JL immunized group. * *p* < 0.05, *** *p* < 0.005 analyzed by unpaired *t*-test.

**Figure 4 vaccines-11-00106-f004:**
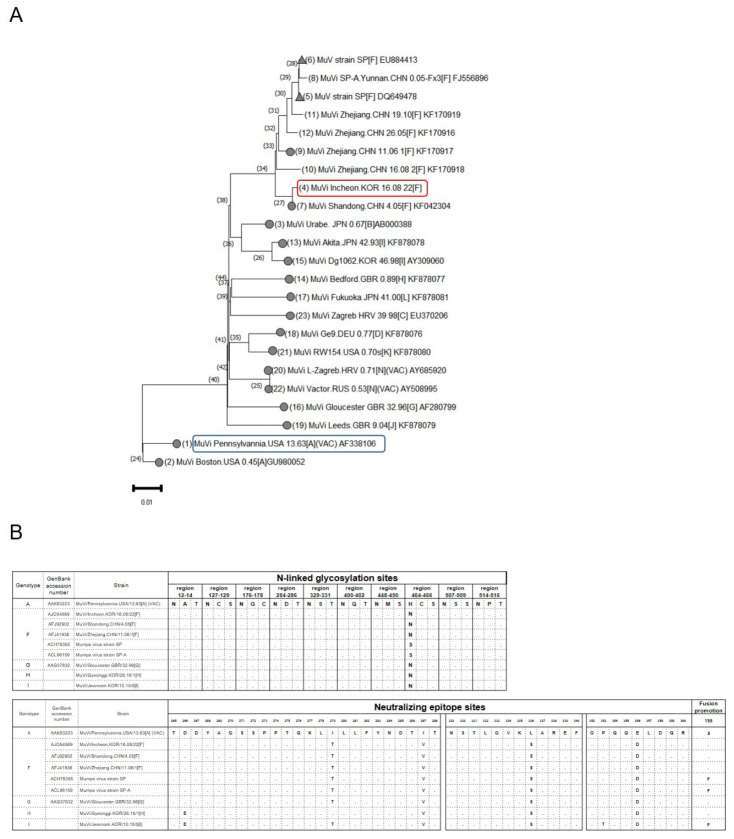
Phylogenetic analysis. (**A**) Whole genome sequencing (WGS) phylogenetic tree. The WGS phylogenetic tree was constructed using the nearly complete genomes (15,349 nucleotides) of wild-type Korean MuV isolates, strain SP, which has been previously studied as an attenuated vaccine labeled with triangles, and 16 WHO reference viruses labeled with circles in NCBI’s GenBank sequence database. The blue box is the current vaccine strain and red box is the inactivated F genotype mumps vaccine (F10) which is the candidate developed in this study. (**B**) N-linked glycosylation sites and neutralizing epitope sites of HN protein sequences were compared between the wild-type and vaccine strains (MuVi. Pennsylvania.USA.13.63) MuV. Fusion promotion site amino acid 195 of the F protein in the wild-type and vaccine strain MuV.

## Data Availability

The data that support the findings of this study are available from the corresponding author upon reasonable request.

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
