# Peer review of "Improved Immunogenicity of the Inactivated F Genotype Mumps Vaccine against Diverse Circulating Mumps Viruses in Mice"

_vaccines, 2023, doi:10.3390/vaccines11010106_

Round 1

Reviewer 1 Report (Previous Reviewer 2)

The authors successfully responded to the reviewer's comments and updated the manuscript as well. 

Author Response

The authors successfully responded to the reviewer's comments and updated the manuscript as well.

Response: Thank you for appreciating our work and providing helpful comments and suggestions, which helped us to improve the quality of our manuscript.

Reviewer 2 Report (New Reviewer)

The Manuscript titled “Improved immunogenicity of the inactivated F genotype mumps vaccine against diverse circulating mumps viruses in mice” by Won et al., has been reviewed. In this study, authors have evaluated the vaccine potential of formalin-inactivated F genotype against Mumps. The candidate vaccine imparted higher cross-neutralizing antibody titers against genotypes F, H, and G as well as significantly high response of Th1 cytokines i.e., IFN-γ, TNF-α, and IL-2 in comparison to Jeryl Lynn (JL) vaccine. The authors concluded that the inactivated F genotype mumps vaccine imparted higher immunogenicity than the JL vaccine against diverse circulating MuVs.

The authors have evaluated the vaccine potential of formalin-inactivated F genotype in BALB/c mice. The study lacks one of the most important experiments as the authors could not complete challenge studies. However, the authors have evaluated the induced humoral and cellular immune response in formalin-inactivated F genotype vaccinated BALB/c mice.

I would like to suggest few corrections/edits before publishing this study.

1.     Authors should mention the proper animal protocols No. (Protocol # and registration #) approved by Institutional Animal Ethics Committee (IAEC) and Institutional Biosafety committee (IBSC) as per the institutional norms for experiments conducted using mice in this study.

2.     Authors should clearly mention in Materials and Methods section, how many mice (N =?) were taken for each group i.e., PBS, JL, F, F10, G and G10 and how many times experiments were repeated (if repeated)?

3.     Redesign the figure 1A and mention all the activities upto 9 weeks after prime as authors have collected the sera at 9W post prime from vaccinated mice.

4.     Under the heading Enzyme-linked immunosorbent assay (ELISA), rewrite the sentence “MuV mixtures (1× 105 ffu/mL of each genotype A, F, H, I, and G) were cultured overnight in 96-well plates (Line 174)”, in this sentence please mention how much volume of MuV mixtures (1× 105 ffu/mL) was added to per/well of ELISA plate. Replace the word ‘cultured’ with coated.

5.     Under the heading “Cytokine quantification (Line 208)”, sentence “The levels of cytokine secreted into the supernatants were quantified” should be rewritten with proper explanation. What is this supernatant? From where the supernatant was collected?

6.     Authors have examined only CD3 levels in the spleen however it has been claimed that the activation of cytotoxic T cells and T helper cells were examined. As shown in Figure 2C, increased CD3 levels were observed in F10 immunized mice compared with JL immunized mice. CD3 is expressed on all T cells and the most reliable pan T cell marker, authors have not seen activation of CD4+ and CD8+ T cells.

7.     A histological analysis of the spleen revealed no significant differences among all groups after vaccination (Line 288). Authors should clearly mention what differences were investigated in the spleen of vaccinated mice and what was their purpose to investigate it?

Author Response

  1. Authors should mention the proper animal protocols No. (Protocol # and registration #) approved by Institutional Animal Ethics Committee (IAEC) and Institutional Biosafety committee (IBSC) as per the institutional norms for experiments conducted using mice in this study.

Response: In accordance with your comment, we have added the numbers (page 6, line 252-257).

  1. Authors should clearly mention in Materials and Methods section, how many mice (N =?) were taken for each group i.e., PBS, JL, F, F10, G and G10 and how many times experiments were repeated (if repeated)?

Response: Following your suggestion, we have added a detailed information in Material and Methods section (page 4, line 147-149). The animal experiments were conducted with a minimum number of animals in accordance with animal experiment ethics, proceeded on an individual basis, and were not repeated.

  1. Redesign the figure 1A and mention all the activities upto 9 weeks after prime as authors have collected the sera at 9W post prime from vaccinated mice.

Response: FIG. 1A is an experiment with blood drawn 3 weeks’ post-boost. It was conducted 6 weeks after priming. To confirm the longevity, we conducted a separate experiment, in two sets, one set examined 3 weeks’ post-boost in other words 6 weeks after priming, and a second set examined 6 weeks post-boost, in other words 9 weeks after priming in Fig 3.

  1. Under the heading Enzyme-linked immunosorbent assay (ELISA), rewrite the sentence “MuV mixtures (1× 105 ffu/mL of each genotype A, F, H, I, and G) were cultured overnight in 96-well plates (Line 174)”, in this sentence please mention how much volume of MuV mixtures (1× 105 ffu/mL) was added to per/well of ELISA plate. Replace the word ‘cultured’ with coated.

Response: In accordance with your comment, we have revised the sentence (page 5, line 176-177).

  1. Under the heading “Cytokine quantification (Line 208)”, sentence “The levels of cytokine secreted into the supernatants were quantified” should be rewritten with proper explanation. What is this supernatant? From where the supernatant was collected?

Response: Following the reviewer’s comment, we have rewritten the sentence. (page 5, line 214, 216-222).

  1. Authors have examined only CD3 levels in the spleen however it has been claimed that the activation of cytotoxic T cells and T helper cells were examined. As shown in Figure 2C, increased CD3 levels were observed in F10 immunized mice compared with JL immunized mice. CD3 is expressed on all T cells and the most reliable pan T cell marker, authors have not seen activation of CD4+ and CD8+ T cells.

Response: We have confirmed CD3 expression, not activity. Figure 2 shows the results confirming the expression of CD3 in a group of mice immunized with the inactivated F10 vaccine, which was slightly higher than that of the JL vaccine. In agreement with your opinion, CD3 acts as a marker for pan-T cells, the staining area did not prominently increase.

However, it was confirmed that CD3 involved in activating both CD4+ and CD8+ was expressed by the F10 vaccine (page 8, line 305-306, 308).

  1. A histological analysis of the spleen revealed no significant differences among all groups after vaccination (Line 288). Authors should clearly mention what differences were investigated in the spleen of vaccinated mice and what was their purpose to investigate it?

Response: In accordance with your comment, we have updated data and the figure legend (page 8, line 305-306, 308). The total T cells within the spleen tissue were evaluated. The results showed that there were significant differences between JL and F10 immunized mice groups in T cells. The increase in T cells was confirmed as CD3, which is a representative marker, and as shown in Fig. 2, F10 immunized mice showed higher CD3 expression than JL immunized mice, suggesting that a high immune response occurred. Thus, it induces strong Th1 cytokines responses, IFN-γ, TNF-α, and IL-2, of CD4+ T cells.

This manuscript is a resubmission of an earlier submission. The following is a list of the peer review reports and author responses from that submission.

Round 1

Reviewer 1 Report

The authors present data on Improved immunogenicity of the inactivated F genotype mumps vaccine against diverse circulating mumps viruses in mice.  There are several major issues that need addressing:

1. The manuscript really needs a thorough revision to make it much clearer what it is describing - it took me some time to work out.

2. There are many areas where the use of English and scientific presentation falls below the minimal level required for publication. Overall a Scientific English revise is essential for the whole text.

3. I really do not understand the data presented in figures and tables

4. The quality of images and tables is very, very poor.

5. The introduction does not provide sufficient background and does not include all relevant references.

6. Not all references are related to research.

7. The research design is not suitable and the methods are not adequately described.

8. The results are not presented clearly and therein lies the big problem with this manuscript.

Author Response

The authors present data on Improved immunogenicity of the inactivated F genotype mumps vaccine against diverse circulating mumps viruses in mice. There are several major issues that need addressing:

Major points

  1. The manuscript really needs a thorough revision to make it much clearer what it is describing - it took me some time to work out.

Response: Thank you for bringing this to our attention. Following your suggestions, we have thoroughly checked the entire manuscript for flow, structure, and order of the sections.

  1. There are many areas where the use of English and scientific presentation falls below the minimal level required for publication. Overall, a Scientific English revise is essential for the whole text.

Response: We have thoroughly checked the entire manuscript for grammatical and language related errors and have made corrections accordingly. Moreover, the manuscript was proofread by a professional English language editor to ensure that there is no language related errors.

  1. I really do not understand the data presented in figures and tables

Response: In accordance with your comment, we have changed sentences and the data presented in figures.

For better understanding, the overall results are summarized below.

To develop mumps vaccine with better immunogenicity, we identified diverse inactivated mumps vaccines. Figure 1 and Supplementary Figure1 showed that inactivated mumps vaccines made up of A, F, H, I, and G genotype virus were prepared and tested for humoral immunity. The F genotype inactivated vaccine has the highest neutralization antibody titer against F, H, and G genotypes of mumps virus.

Figure 2 indicated that the F genotype inactivated vaccine induced cell-mediated immunity e.g., Th1 cytokine responses.

We tested the longevity of neutralization antibody titers with inactivated mumps vaccine candidates shown in Figure 3. Figure 1C shows the F genotype inactivated vaccine has the highest neutralization antibody titers against F, H, and G genotypes of mumps virus despite concerns that inactivated vaccines are less durable than other vaccine platforms.

In Figure 4, whole genome sequence differences between current mumps vaccine (Jeryl Lynn strain) and newly identified F genotype mumps strain (F10) used in this study and between F genotype attenuated vaccines (Mumps virus strain SP) have already been developed and F10 were analyzed. N-glycosylation site at amino acid (aa) 464 is absent in the vaccine strain. the fusion promotion site at aa 195 in the F protein of MuV, MuVi/Incheon.KOR/16.08/22[F] is the same as that of the JL vaccine strain, whereas that in the strain SP has a mutation of S to F.

  1. The quality of images and tables is very, very poor.

Response: Following the reviewer’s comment, we have increased resolution to 600 dpi

  1. The introduction does not provide sufficient background and does not include all relevant references.

Response: In accordance with your comment, we have removed unnecessary sentences and revised Introduction and References (page 3 and 17)

  1. Not all references are related to research.

Response: In accordance with your comment, we have removed irrelevant references. (page 17)  

  1. The research design is not suitable and the methods are not adequately described.

Response: Following the reviewer’s comment, the order of the data was changed and revised. The experimental method was revised using the new reference. (page 6 line 131, page 9 line 185)

  1. The results are not presented clearly and therein lies the big problem with this manuscript.

Response: Following the reviewer’s comment, we have revised the overall results and the entire manuscript. (page 11 line 252- page 12 line 259 and line 272-2)

Reviewer 2 Report

1.       There are no references in the material and methods sections. Please include the relevant references.

2.       Is there any specific reason, that only female mice have been used in the study? Please justify and discuss.

3.       The quality of the figures is not at publication standard. Also, there should be homogeneity in figures, such as colors and representation style. Please update.

4.       Figure1. Why is only the intramuscular route of immunization used? Please justify and discuss. (B) Why are first immunization and second immunization specifically mentioned below the group zero and three? What is group zero? (C) All labels should be detailed in the figure legend. Please update.

5.       What about statistical significance in Figure 2A? What are genotypes A, F, H, I, and G in figure 2B? Please detail in the figure legend.

6.       Figure4. (A) Why are first immunization and second immunization specifically mentioned below the group zero and three? What are groups 0, 7, 8, and 9? What about statistical significance in Figure 4B?

7.       Figure5. What type of phylogenetic tree is this? Please include the details in the figure legend. Also, please include a root in the tree.

8.       Please make a separate section for conclusions. Also, include a separate section for limitations to the present study.

Author Response

  1. There are no references in the material and methods sections. Please include the relevant references.

Response: In accordance with your comment, we have added relevant references in the Material and Methods. (page 6 line 131, page 9 line 185)

  1. Is there any specific reason, that only female mice have been used in the study? Please justify and discuss.

Response: As females have been reported to exhibit more injurious reactions than male using the MMR vaccine, therefore, we studied female mice. Please see the references below

Khalil, M. K., et al. "Effect of gender on reporting of MMR adverse events in Saudi Arabia." EMHJ-Eastern Mediterranean Health Journal, 9 (1-2), 152-158, 2003 (2003).

Ruggieri, Anna, et al. "The influence of sex and gender on immunity, infection and vaccination." Annali dell'Istituto superiore di sanita 52.2 (2016): 198-204.

  1. The quality of the figures is not at publication standard. Also, there should be homogeneity in figures, such as colors and representation style. Please update.

Response: Following the reviewer’s comment, we have unified the data and improved image quality.

  1. Figure1. Why is only the intramuscular route of immunization used? Please justify and discuss.

Response: IM is commonly used route for licensed vaccines due to easy accessibility and safety and has high immunogenicity in case of influenza virus vaccine. Thus, we also conducted immunization in mice using the IM route.

Cook, Ian F., et al. "Reactogenicity and immunogenicity of an inactivated influenza vaccine administered by intramuscular or subcutaneous injection in elderly adults." Vaccine 24.13 (2006): 2395-2402.

Harris, Katie, et al. "Intramuscular immunization of mice with live influenza virus is more immunogenic and offers greater protection than immunization with inactivated virus." Virology Journal 8.1 (2011): 1-11.

Cook, Ian F. "Evidence based route of administration of vaccines." Human vaccines 4.1 (2008): 67-73.

(B) Why are first immunization and second immunization specifically mentioned below the group zero and three? What is group zero? (C) All labels should be detailed in the figure legend. Please update.

Response: In accordance with your comment, we have updated data and the figure legend. The manuscript (vaccines0205489) was revised and the initial immunization started at zero. The number means the week. In addition, we updated description of all figures (page 22, line 517-522 and page 26).

  1. What about statistical significance in Figure 2A? What are genotypes A, F, H, I, and G in figure 2B? Please detail in the figure legend.

Response: In accordance with your comment, we have updated data and the Figure legend (page 22, line 524-525 and page 23, line 527-528).

  1. Figure 4. (A) Why are first immunization and second immunization specifically mentioned below the group zero and three? What are groups 0, 7, 8, and 9? What about statistical significance in Figure 4B?

Response: As immunizations were performed according to the schedule in Fig. 1A, we have deleted the diagram with the same experimental schedule. All other experiments collected blood and spleen from the mice at 6 weeks. Fig. 3 were sacrificed into 2 groups for 6 and 9 weeks, respectively, for neutralizing antibody persistence experiments. We moved Fig. 4B to Supplementary Fig. 4 because the data of total IgG data in Fig. 4B is similar to Fig 1B. We have updated data, the figure legend, and statistical analysis results. (page 23, line 540-542 and page 25, line 580-581)

  1. Figure 5. What type of phylogenetic tree is this? Please include the details in the figure legend. Also, please include a root in the tree.

Response: Following the reviewer’s comment, we reconstructed a phylogenetic tree using MEGA 11 version including the root, which is the current vaccine strain. Due to small sample size, MuV whole-genome sequences from WHO references strains, some F genotype mumps virus sequences from the NCBI database and Korean MuV isolates were analyzed using the neighbor-joining method with 1,000 bootstrap replicates. We have added detailed information in Material and Methods section. (page 10, line 229-page 11, line 233, page 23, line 544-page 24, line 550):

  1. Please make a separate section for conclusions. Also, include a separate section for limitations to the present study.

Response: Following the reviewer’s comment, we have made a separate section for conclusion and limitation. (page 15, line 353 and page 16, line 358)